# The COVID-19 related stress and social network addiction among Chinese college students: A moderated mediation model

**Ziao Hu[1], Yangli Zhu[2], Jun Li[1]\*, Jiafu Liu[3], Maozheng Fu[1]**

**1** School of Finance and Economics, Hainan Vocational University of Science and Technology, Haikou, China, **2** Student Affairs Office, Yunnan Normal University, Kunming, China, **3** Guizhou Education University, Guiyang, China

\* lijun.edu.ma@foxmail.com

**Data Availability Statement:** All data files are available from the OSF database (accession number (s) DOI 10.17605/OSF.IO/ZR5YK).

## Abstract

Based on social cognitive theory and gender differences, this study verified a moderated mediation model to explore the relationship between the COVID-19 related stress (CRS) and social network addiction (SNA) and evaluate the mediating role of fear of missing out (FoMO) and the moderating role of gender. A questionnaire survey was conducted, including 702 Chinese university students.This study used PROCESS to test the hypothesis model.The results showed that the CRS significantly and positively affected the SNA of college students and FoMO played a complementary mediating role. Moreover, the analysis of the moderated mediation model showed that gender moderated the relationship between FoMO and SNA; the effect of FoMO was stronger on the SNA of male college students than that of females. The results not only enhanced our understanding of the internal influencing mechanism of the relationship between CRS and SNA but also considered gender differences. In addition, some suggestions were proposed.

## Introduction

Social networking sites (SNSs) are virtual communities for building new interpersonal communication, where users can create personal public profiles, interact with friends, and perform activities based on shared interests [1, 2]. SNSs are attractive entertainment platforms [3] that are becoming increasingly popular worldwide. Particularly for adolescents and students [1, 4–7], SNSs are the primary means of interaction in their free time [8]. According to reported statistics, Facebook alone, one of the leading SNS today, has more than 2 billion users [9]. Users can present themselves and rebuild their social identities by creating their profiles on SNSs [3]. Meanwhile, because SNSs provide private environments free from parental supervision, adolescents can easily interact with their friends and peers happily anytime and anywhere. These advantages attract youngsters to use social networks frequently [10].

Although using SNSs can bring convenience to individuals and society, it may also have adverse effects, such as the problem of social network addiction (SNA) [4]. Previous studies have reported that SNA can lead to harmful psychological problems such as anxiety, depression, loneliness [4], and low self-esteem [11]. Moreover, it can cause more negative physical

**Funding:** The author(s) received no specific funding for this work.

**Competing interests:** The authors have declared that no competing interests exist.

**Abbreviations:** CRS, COVID-19 related stress; FoMO, fear of missing out; SNA, social network addiction.

problems such as eye diseases [12], excessive sitting [1, 13]. SNA severely damages the physical and mental health of adolescents and destroys their well-being in life [14, 15], and in severe cases, it may lead to suicide [16]. The degree of social network addiction is on the rise [17], which has attracted the continuous attention of scholars [2, 4, 14–16, 18–26]. Adolescents are at high risk of social network addiction [17, 25, 27]. Compared with other countries, SNA is the most prominent among Chinese college students [28], which has also attracted considerable attention from the Chinese government and researchers [29–31]. Recent studies have shown that SNA among university students worsened during the COVID-19 pandemic [32]; therefore, the factors that affect SNA among university students during COVID-19 pandemic need to be immediately identified.

Previous researchers have generally concluded that factors influencing SNA are multifaceted, such as social attachment [33], lack of self-control [34], emotions [4], technology [33], enjoyment [35], and stress [4, 36]. Moreover, past studies have reported that stress under certain conditions is one of the driving factors of SNA [4, 36]. As COVID-19 has become health, economic, and social emergency [37] and a unique disaster [38], great changes have taken place in every aspect of people's life [39]. These dramatic changes have added to the stress levels of college students in their studies and lives [39, 40]. Numerous studies have confirmed stress to be a significant predictor of SNA [26, 41–43]. Moreover, previous studies have reported that fear of missing out (FoMO), a negative personal emotion [44] amplifying favorable experiences in social networks [1, 45], is strongly associated with SNA [46, 47] and drives the growth of social network addictive tendencies [48]. Therefore, this study suggested that the COVID-19 related stress (CRS) and FoMO may directly affect SNA of college students.

Meanwhile, recent studies have shown a significant positive correlation between stress and FoMO [46, 49], and stress significantly positively predicts FoMO [49]. In addition, FoMO is often considered a mediator in studies on negative social network behaviors [49, 50] and has become a crucial variable for research attention [47]. Therefore, this study inferred that FoMO may mediate between CRS and SNA among university students. Furthermore, general and persistent differences exist between genders in perceptions of the external environment (cognitive), personal factors (psychological and personality), and behavioral factors (ability, functioning, and role) [51]. Previous related studies that included FoMO [52], addiction problems [53], and stress [54, 55] have reported gender as a moderating variable. Therefore, this study explored whether gender plays a moderating role in the effect of CRS on the SNA of college students via FoMO.

Although several previous studies exist on the impact of stress on SNA. But in fact, the COVID-19 caused massive school closures [56]; the postponement or cancellation of most activities and events, and a lot of formal and informal interactions to online platforms; the COVID-19 significant changes in academic and life patterns of college students and potentially caused social isolation. Therefore, the COVID-19 related stress is a new phenomenon responding in a specific context, and this stress has been proved to have a very significant impact on the study and life of college students, which needs the attention of researchers [39]. At the same time, the measurement of stress caused by sudden major public health events should be different from the general measurement of stress, and most of the previous studies have adopted the general sense of stress measurement tools for assessment [34], which may lead to certain limitations in the research results. Therefore, this study adopted a targeted questionnaire developed by Zurlo et al. [39] to specifically evaluate the pressure related to COVID-19 among college students to further expand the previous research. In addition, while previous studies have explored pairwise structural relationships between stress, FoMO, and SNA in general, the relationship between the three has not been discussed in the same hypothetical model, and the effects of gender differences have not been considered on this basis.

On the basis of social cognitive theory (SCT), wherein the individual, environment, and behavior interact [57]. This study regarded the CRS as an environmental factor and FoMO as a personal factor to explore their potential influence mechanism on SNA of college students. Moreover, this study took gender differences into account, constructing a moderated mediation model to discuss these comprehensive effects. By trying to expand the social cognitive theory and the above studies through the research results, we can not only enrich our understanding of how closely these variables were associated but also provide a valuable reference for colleges and universities to help students correctly understand SNA and prevent its further deterioration. Making a contribution in these senses is the goal of this study.

## Literature review

### Social cognitive theory

SCT, proposed by Bandura [57], believes that the environmental and individual cognition factors have an impact on individual behavior. SCT has been used to study SNA in various countries and groups [58–60]. Specifically, Chen et al. [58] studied the effect of COVID-19 victimization experience (environmental factor) on mobile phone addiction of college students (behavioral factor) through future anxiety (personal factor) based on SCT. Moreover, the results showed that future anxiety plays a fully mediating role in the relationship between the effect of COVID-19 victimization experience and mobile phone addiction of college students. Wu et al. [59] conducted a study on 277 Chinese Macao youth using the SCT model as a framework considering outcome expectancies as an environmental factor, Internet self-efficacy as a personal factor, and SNA as a behavioral factor; the results revealed that both outcome expectancy and online self-efficacy positively predicted SNA. Yu et al. [60] investigated 395 Chinese people to explore the effects of low optimism and loneliness on SNA of college students based on SCT; the results showed that low optimism is an indirect risk factor for SNA, whereas loneliness is a direct risk factor for SNA.

Therefore, on the basis of SCT, this study considered the CRS as an environmental factor, FoMO as an individual factor and SNA as a behavioral factor. In addition, gender was included as a moderating variable to construct a moderated mediation model to explore the effect of CRS on SNA through FoMO and the mediating role of gender.

### CRS and SNA

Similar to SARS and H1N1, COVID-19, as a public health emergency with extremely fast transmission speed and high infection rate, has been prevalent all over the world in recent years [61]. Studies have suggested that such public health emergencies can spread stress, lead to psychotic symptoms such as panic and even suicide [62]. The COVID-19 has dramatically changed the lives of college students [39]. Studies have shown that college students are one of the groups that were most affected by COVID-19 [63]. The psychological stress caused by COVID-19 among college students is enormous [64] and multifaceted [39]. A recent study on medical undergraduates revealed that remote online examinations adopted during the pandemic were more stressful than on-campus examinations [65]. Moreover, data from a longitudinal study by Hakami et al. [40] supported the findings of Elsalem et al. [65], drastic changes such as social restrictions during the COVID-19 pandemic are putting extra stress on college students [40]. To further explain the stressors caused by COVID-19, Zurlo et al. [39] investigated more than 500 European university students during the Covid-19 pandemic. The results indicated that the stressors of the COVID-19 in university students were "interpersonal and academic life," "social isolation," and "fear of contagion," and a seven-item COVID-19 Student

Stress Questionnaire (CSSQ) was developed to evaluate the stress of students related to COVID-19 [39].

SNS can be used as a method to relieve stress and is the easiest and convenient method to reduce stress [66]. In response to daily stressors, individuals may use SNS more frequently, leading to SNA [36, 67, 68]. Therefore, stress can be a factor in SNA [36]. Recent studies have indicated that a significant positive correlation between stress and SNA [24], and higher levels of perceived stress are associated with a greater risk for SNA [26]. Adolescent students who experience stress in a learning environment are most likely to have SNA [69]. Moreover, because the COVID-19 pandemic had put too much pressure on college students in various aspects [39, 64], it may have led them to overuse social networks [64], thus leading to SNA [1, 36]. In view of the above discussion, this study speculates that college students in the context of COVID-19 pandemic may become addicted to social networks due to COVID-19 related stress. Therefore, the first aim of this study is to explore whether COVID-19 related stress has a significant positive effect on the SNA of Chinese college students.

### The mediating role of FoMO

FoMO is a state of anxiety that one will miss something [70]. It is a fear that others may be having beneficial experiences of which one is not a part, a desire to be continuously connected to others, and a negative emotion caused by unmet social relationship needs [44]. Studies suggest that individuals experience greater levels of FoMO when their social needs are unmet [71, 72]. The decline of individual social activities during the pandemic [73], may likely to lead to inadequate socialization and thus increase the FoMO. Recent studies have shown a positive association between stress and FoMO [46, 49]. Fabris et al. [46] surveyed 472 European adolescents and found that Sensitivity to Stress associated with negative experience on social media, which is neglected by online peers, was positively associated with FoMO. Yang et al. [49] investigated 2276 Chinese college students and revealed that stress was not only positively associated with FoMO but also positively predicted FoMO, and stress could also further affect problematic mobile phone use through FoMO.

Studies have shown that FoMO is strongly associated with SNA [46, 47]. Online social networking creates a platform to socially connect and engage, attracting people with FoMO [74]. Thus, SNSs are often used to satisfy FoMO through online social activities to alleviate negative emotions [75]. Such negative emotions may cause people to dedicate more energy to SNSs at the expense of real life and cause more real-life problems such as interpersonal relationships and work [1] and thus generate new negative emotions in return. In this vicious cycle, the psychological dependence on SNSs is further increased [17], eventually causing SNA [1, 36]. Findings show that people with high FoMO are more active in responding to messages on SNSs [76]. Fabris et al. [46] suggested that FoMO was a promoting factor for social media addiction. Moore & Craciun [48] further confirmed the significant predictive effect of FoMO on social media addictive tendencies and indicated that FoMO was the most critical driver of growing addictive tendencies in social networks.

Previous researchers have found that FoMO often acts as a mediator in studies predicting the behavior of social network use [44, 47, 50, 77]. For instance, Beyens et al. [77] noted that FoMO fully mediated the effect of the need to belong on Facebook. In Liu & Ma's [50] empirical research model, FoMO mediated the relationship between anxious attachment and SNS addiction. In summary, this study suggests that CRS may significantly affect SNA through FoMO. The second aim of this study is to investigate whether FoMO has a mediating role in the influence relationship of CRS on SNA of Chinese college students.

## The moderating role of gender

Previous researchers have revealed general and persistent differences between genders in the cognitive, psychological, personality, and behavioral aspects of the external environment [51].

First, gender is an essential determinant of psychological stress perception [78] and significant gender differences exist in the perception of stress [79]. Recent studies have shown significant gender differences in perceived stress among different groups such as surgeons, police officers, high school students, and college students [79–82].

Second, the findings of gender differences in FoMO are controversial. On the one hand, an online study conducted in Germany observed differences in FoMO by age and not by gender [83]. No statistically significant gender differences were observed in the relationship between FoMO and social network use during the COVID-19 pandemic [84]. On the other hand, a cross-sectional study conducted by Li et al. [53] found significant gender differences in FoMO among Chinese college students, and Munawar et al. [52] showed significant gender differences in FoMO in 324 respondents, wherein women were more sensitive to FoMO than men.

Third, previous studies have pointed out that addictive behaviors may be associated with gender differences [85, 86], and significant differences exist in SNA by gender [87–89]. Specifically, a meta-analysis showed significant gender differences in SNA [88]. A study conducted in Spain with university students found similar results supporting gender differences in SNA [85].

Furthermore, previous studies on stress [54, 55], FoMO [52], and addiction [53] have introduced gender differences as moderating variables for discussion. Therefore, this study focuses on gender differences, and the third aim is to explore whether gender has a moderating role on the mediating model of CRS influencing college students' SNA through FoMO.

## The present study

Based on the above theories and empirical studies, this study constructs a moderated mediation model (Fig 1). Compared with simple mediation or moderation models, integrated moderated mediation model provides a deeper understanding of the potential influence mechanism of college students' SNA [90]. Corresponding to the above research objectives, the following research hypotheses are presented in this study:

Hypothesis 1 (H1): CRS has a significant positive effect on Chinese college students' SNA.

Hypothesis 2 (H2): FoMO has a mediating role in the influence relationship of CRS on SNA of Chinese college students.

Hypothesis 3 (H3): Gender has a moderating role in the direct and indirect effects of CRS on SNA through FoMO.

H3a: Gender has a moderating role between CRS and FoMO.

H3b: Gender has a moderating role between FoMO and SNA.

H3c: Gender has a moderating role between CRS and SNA.

## Method

### Ethics approval

This study was approved and ethically reviewed by the Academic Ethics Committee of Hainan Vocational University of Science and Technology (HKD-2022-25). The Declaration of Helsinki and ethical standards were followed [91]. The corresponding author was designated to be responsible for data extraction and analysis.

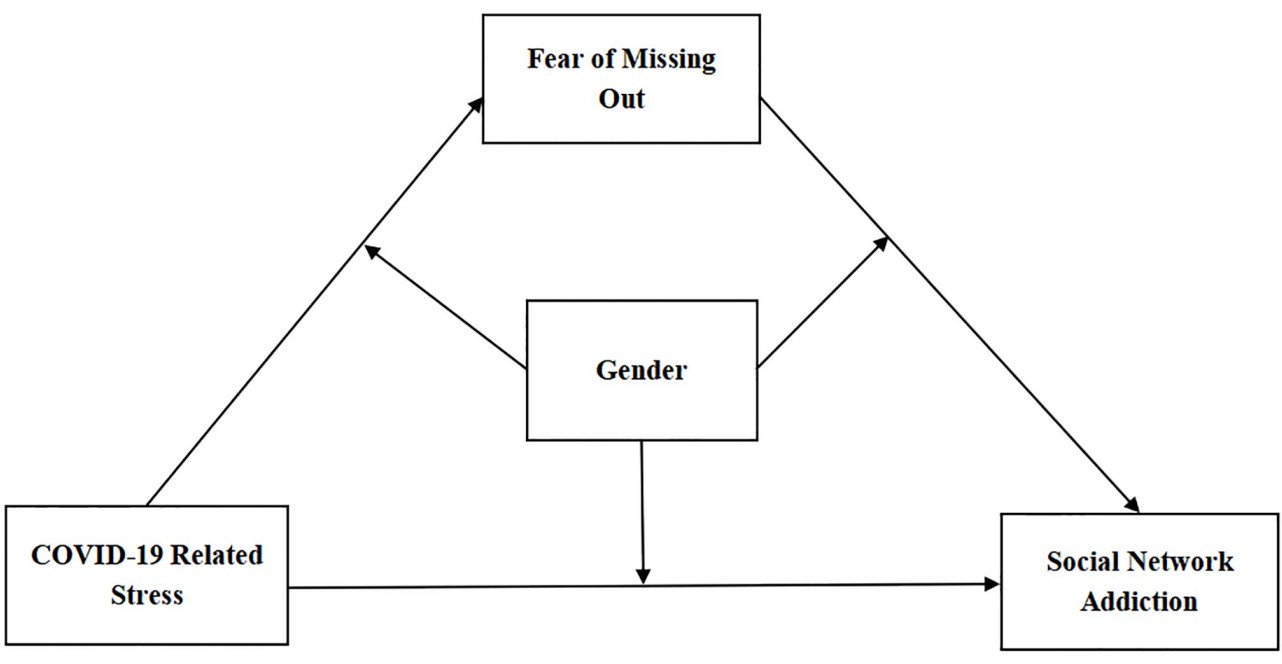

**Fig 1. Moderated mediation model.**

## Participants and procedure

In this study, purposive sampling was used to recruit college students at a university in Southern China. The criteria for recruiting participants were college students who voluntarily wanted to participate should be recruited. First of all, professional training was provided to teachers who would distribute the questionnaire. The questionnaire and items were explained to make them understand the purpose of the study. Second, the participants were informed of the anonymous submission of the questionnaires, the study's purpose, and the confidentiality agreement. After obtaining consent from participants, questionnaires were distributed through the online questionnaire platform Questionnaire Star (www.wjx.cn). By scanning a two-dimensional barcode, participants could complete the survey with the help of their teachers. They could refuse or withdraw from the study anytime before submission.

According to the formula for calculating the sample size by Israel [92], the official sample size of this study should not be less than 652. In our case, 777 questionnaires were distributed, 702 valid questionnaires were returned (75 invalid questionnaires were excluded). The effective rate was 90.35%. There were 170 (24.2%) male and 532 (75.8%) female students; 236 (33.6%) only children and 466 (66.4%) non-only children; 317 (45.2%) first year university students, 160 (22.8%) second year students, 152 (21.7%) third year students, and 30 (4.3%) fourth year students; and 43 master's students (6.1%). The gender imbalance in the sample is because the sampled university was a normal university, including a larger proportion of female students in the natural sample composition [93].

## Instruments

**CRS.** This study used the COVID-19 Student Stress Questionnaire (CSSQ),which was developed by Zurlo et al. [39] The questionnaire was tested for good reliability among college students [39]. Three dimensions of the scale were relationships and academic life, isolation

and fear of contagion. It was a five-point Likert scale ranging from 1 (not at all stressful) to 5 (extremely stressful), with higher scores indicating more stress from COVID-19.

**SNA.**  This study used the SNA scale for adolescents, which was developed by Wang et al. [94] to measure SNA among college students. It was a five-point Likert unidimensional scale, ranging from 1 (not at all true of me) to 5 (extremely true of me), with higher scores indicating higher levels of SNA.

**FoMO.**  This study used the FoMO scale, which was developed by Przybylski et al. [44] to measure FoMO among college students. It was a five-point Likert unidimensional scale, ranging from 1 (not at all true of me) to 5 (extremely true of me), with higher scores indicating higher levels of FoMO.

## Statistical analysis

In this study, first, we first conducted the reliability and validity tests of each measurement instrument. The reliability was reflected by Cronbach's α, and when Cronbach's α was greater than 0.7, the reliability was better [95]. Confirmatory factor analysis (CFA) was used to test validity and model fit. Standardized Factor Loading (SFL), Composite Reliability (CR) and Average Variance Extracted (AVE) of each measured model were tested, according to the suggestion of Cheung & Wang [96], SFL>0.5, CR> 0.7 and AVE>0.5, indicated that the convergent validity of the measurement model is good. Considering the sensitivity of chi-square value to large sample sizes, chi-square value was not reported in this study when the fitness of the measurement model was reported. According to Hu & Bentler [97], when the sample size was large, the chi-square values tended to reach significance and other fit indexes could be referred to. In this study, the following model fit indexes were reported: RMR<0.08; GFI>0.85; CFI>0.85; NFI>0.85; TLI>0.80; IFI>0.85 and PNFI>0.5 [98]. If the above criteria are satisfied, the measurement model fitness is acceptable. In addition, the discriminant validity of each potential variables needed to be accounted for. The square root of AVE was performed to assess the discriminant validity of each dimension of the measurement model. The criterion was that the square root of the AVE of each dimension should be greater than the correlation coefficient of each dimension [99].

Secondly, since the questionnaire of this study was collected by online self-report, it was necessary to test the Common Method Variance (CMV) before data analysis. Harman's One-Factor test was used to test the CMV. The Kaiser-Meyer-Olkin (KMO) should be greater than 0.8, with the Bartlett test of sphericity reaching significant ($p < 0.001$) and the explanatory power of the first factor no exceeding the critical value of 50% [100]. If the above indexes could meet the criteria simultaneously, indicating that the CMV problem did not affect the findings.

Thirdly, descriptive statistics and correlation analysis were performed on the study variables. The descriptive statistics reflected the mean and standard deviation of each variables. Pearson's correlation coefficient reflected the correlation between the study variables, and the correlation coefficient was less than 0.7 [96], indicating that all variables had no collinearity problem, and regression analysis could be performed.

Fourth, Model 4 of PROCESS was used to test the mediation role of FoMO, with the CRS as the independent variable, FoMO as the mediating variable, and SNA as the dependent variable. Furthermore, the mediation models should be classified into four types: the full mediation, the partial mediation (the complementary partial mediation and the competitive partial mediation), only direct effect and no effect [101, 102]. The type of mediation model would be reported in this study. Model 59 of PROCESS added gender as a moderator variable to validate the mediating role of moderation. Meanwhile, the bootstrap confidence interval (CI) was set

to 95%, and the number of samples was set to 5,000. The CI cannot contain 0, indicating a significant effect [103].

# Results

## Measurement model

**CRS.** The CFA results indicated that the last item of the scale had factor loadings less than 0.5 and was removed [104]. Standardized factor loading (SFL) for the retained items ranged from 0.787 to 0.874, both greater than 0.5; Composite Reliability (CR) values were 0.890 and 0.861, both greater than 0.7; and Average Variance Extracted (AVE) values were 0.670 and 0.756, both greater than 0.5. These three indices indicated the ideal convergent validity of the measurement model [96]. The fit indices of the measurement model were as follows: RMR = 0.039, GFI = 0.976, CFI = 0.984, NFI = 0.981, TLI = 0.970, IFI = 0.984, and PNFI = 0.523, indicating a good fit of the measurement model [98]. Cronbach's α of the total scale was 0.906 (>0.7) [95].

**SNA.** The results shown that the SFL ranged from 0.694 to 0.806, all values greater than 0.5. The CR value was 0.919, greater than 0.7. The AVE value was 0.587, greater than 0.5. These three indices indicate the ideal convergent validity of the measurement model [96]. The fit indicators of the measurement model were as follows: RMR = 0.046, GFI = 0.905, CFI = 0.928, NFI = 0.923, TLI = 0.900, IFI = 0.929, and PNFI = 0.660, indicating a good fit of the measurement model [98]. Cronbach's α of the total scale was 0.919 (>0.7) [95].

**FoMO.** The CFA results indicated that three items had factor loading less than 0.5 and were removed [104]. The SFL for the retained items were 0.573–0.858, all values greater than 0.5; the CR value was 0.901, greater than 0.7; and the AVE value was 0.571, greater than 0.5. These three indices indicate the ideal convergent validity of the measurement model [96]. The fit indicators of the measurement model were as follows: RMR = 0.063, GFI = 0.851, CFI = 0.877, NFI = 0.873, TLI = 0.815, IFI = 0.877, and PNFI = 0.582, indicating that the fit of the measurement model to the observed data was acceptable [98]. Cronbach's α of the total scale was 0.901(>0.7) [95].

## Discriminant validity

To validate the discriminant validity of all dimensions, we adopted the rigorous testing method of the square root of AVE. As shown in Table 1, the results demonstrated that the square root of AVE in each dimension was greater than the correlation coefficient in each dimension, indicating that the discriminant validity of each scale was good [99].

## CMV test

To validate CMV, we performed Harman's one-factor test. Unrotated factor analysis revealed that KMO was 0.925 (>0.8), and the Bartlett test of sphericity reached significance

**Table 1. Discriminant validity.**

| Dimension | M | SD | Relationships and Academic Life | Isolation | SNA | FoMO |
|---|---|---|---|---|---|---|
| Relationships and Academic Life | 2.562 | 1.077 | *0.819* | | | |
| Isolation | 2.909 | 1.252 | 0.698*** | *0.869* | | |
| SNA | 2.952 | 0.806 | 0.356*** | 0.308*** | *0.766* | |
| FoMO | 2.661 | 0.812 | 0.437*** | 0.377*** | 0.579*** | *0.756* |

Notes: n = 702; the bold and italic numbers are the square root of AVE. the lower numbers are the correlation coefficients of two dimensions.

***p<0.001.

**Table 2. Descriptive statistics and correlation analysis.**

| Variable | M | SD | GENDER | CRS | SNA | Fomo |
|---|---|---|---|---|---|---|
| GENDER | 0.240 | 0.429 | 1 | | | |
| CRS | 2.677 | 1.053 | 0.037 | 1 | | |
| SNA | 2.952 | 0.806 | -0.084* | 0.364*** | 1 | |
| FoMO | 2.661 | 0.812 | -0.047 | 0.447*** | 0.579*** | 1 |

**Note:** n = 702; Gender was treated as a dummy variable;

*p<0.05

***p<0.001.

($p < 0.001$). The analysis revealed three factors, and the explanatory power of the first factor was 42.271%, which did not exceed 50% [100], indicating that the CMV problem was not significant.

## Descriptive statistics and correlation analysis

The descriptive statistics and correlation analysis for the CRS, SNA, FoMO, and gender are shown in Table 2. The correlation analysis showed that CRS was positively and significantly correlated with SNA (r = 0.364, $p < 0.001$) and FoMO (r = 0.447, $p < 0.001$), and FoMO was positively and significantly correlated with SNA (r = 0.579, $p < 0.001$). Meanwhile, gender was negatively and significantly correlated with SNA (r = −0.084, $p < 0.05$), and no significant correlation existed between gender and CRS and FoMO. The absolute values of the correlation coefficients between any two variables in this study were less than 0.7, indicating no collinearity problem [96].

## The mediating role of FoMO

The mediating effect of FoMO was tested using Model 4 of PROCESS. The results are shown in Table 3. After controlling for gender, the CRS significantly and positively predicted SNA in model 1 (B = 0.282, $p < 0.001$), and H1 was supported. In addition, it significantly and positively predicted FoMO in model 2 (B = 0.347, $p < 0.001$). In model 3, after adding FoMO as a mediating variable, FoMO significantly and positively predicted SNA (B = 0.510, $p < 0.001$), and CRS still predicted SNA significantly (B = 0.105, $p < 0.001$), indicating that FoMO had a

**Table 3. Testing the mediation model of FoMO.**

| Variable | Model 1 SNA | | | Model 2 fomo | | | Model 3 SNA | | |
|---|---|---|---|---|---|---|---|---|---|
| | B | SE | 95% CI | B | SE | 95% CI | B | SE | 95% CI |
| GENDER | -0.183** | 0.066 | (-0.317, -0.045) | -0.121 | 0.064 | (-0.248, 0.005) | -0.121* | 0.057 | (-0.228, -0.007) |
| CRS | 0.282*** | 0.027 | (0.219, 0.341) | 0.347*** | 0.026 | (0.285, 0.405) | 0.105*** | 0.026 | (0.047, 0.165) |
| FoMO | | | | | | | 0.510*** | 0.034 | (0.430, 0.586) |
| $R^2$ | 0.142 | | | 0.204 | | | 0.353 | | |
| F | 57.960*** | | | 89.722*** | | | 126.895*** | | |

**Note:** Gender was treated as a dummy variable; B are unstandardized coefficients; SE, standard error; CI, Confidence Interval;

*p<0.05

**p<0.01

***p<0.001.

partial mediating effect in the relationship between CRS and SNA, and H2 was supported. The bias-corrected nonparametric percentile bootstrap method was further used to test the mediating effect of FoMO. The indirect effect value was 0.177, and the 95% CI ranged from 0.136 (LLCI) to 0.220 (ULCI), excluding 0, indicating a mediating effect. The direct effect value was 0.105, and the 95% CI ranged from 0.047 (LLCI) to 0.165 (ULCI), excluding 0, again validating the partial mediating effect of FoMO with the mediating effect accounting for 62.766% of the total effect. Meanwhile, the direct (B = 0.105, p < 0.001) and indirect effects (0.347 × 0.510 = 0.177, p < 0.001) of CRS on SNA were both positive (the direct and indirect effects pointing in the same direction), indicating a complementary partial mediation.

## The moderating role of gender

To test whether gender moderated the direct and indirect relation among the CRS, FoMO, and SNA, Model 59 of PROCESS was used. The results are shown in Table 4 and Fig 2. After controlling for gender, a significant positive predictive effect of CRS on FoMO was observed in model 1 (B = 0.358, p < 0.001), and the interaction term between CRS and gender was not a significant predictor of FoMO (B = −0.036, p > 0.05), indicating that gender has no moderating effect in the relationship between CRS and FoMO, thus H3a was not supported. In model 2, a significant positive predictive effect of CRS was observed on SNA (B = 0.123, p < 0.001), and the interaction of CRS with gender was insignificant in predicting SNA (B = −0.064, p > 0.05), indicating that gender had no moderating effect on the relationship between CRS and SNA, thus H3c was not supported. Meanwhile, a significant positive predictive effect of FoMO on SNA (B = 0.465, p < 0.001) and the interaction term between FoMO and gender on SNA (B = 0.169, p < 0.05) was observed, indicating a moderating role of gender between the FoMO and SNA, thus H3b was supported.

The moderating effect of gender was further validated using the bias-corrected nonparametric percentile bootstrap method. The results again confirmed that gender had a significant moderating effect only in the second half of the mediation model constructed in this study. Particularly, gender moderated the relationship between FoMO and SNA (B = 0.169, p < 0.05, 95% CI = 0.025–0.313). Therefore, the indirect effect of CRS on SNA through FoMO was stronger in male college students (B = 0.241, 95% CI = 0.182–0.299) than females (B = 0.173, 95% CI = 0.128–0.221). To visualize the moderating effect of gender, the moderating effect was plotted (Fig 3). From the simple slope analysis, FoMO had a stronger predictive effect on SNA

**Table 4. Testing the moderated mediation model.**

| Variable | Model 1 fomo | | | Model 2 SNA | | |
|---|---|---|---|---|---|---|
| | B | SE | 95% CI | B | SE | 95% CI |
| GENDER | -0.022 | 0.166 | (-0.301, 0.268) | -0.389* | 0.197 | (-0.736, -0.044) |
| CRS | 0.358*** | 0.031 | (0.296, 0.421) | 0.123*** | 0.031 | (0.065, 0.182) |
| CRS × GENDER | -0.036 | 0.056 | (-0.150, 0.073) | -0.064 | 0.056 | (-0.176, 0.055) |
| FoMO | | | | 0.465*** | 0.040 | (0.385, 0.539) |
| FoMO × Gender | | | | 0.169* | 0.076 | (0.025, 0.313) |
| $R^2$ | 0.205 | | | 0.357 | | |
| F | 59.904*** | | | 77.450*** | | |

**Note:** Gender was treated as a dummy variable;

*p<0.05

***p<0.001.

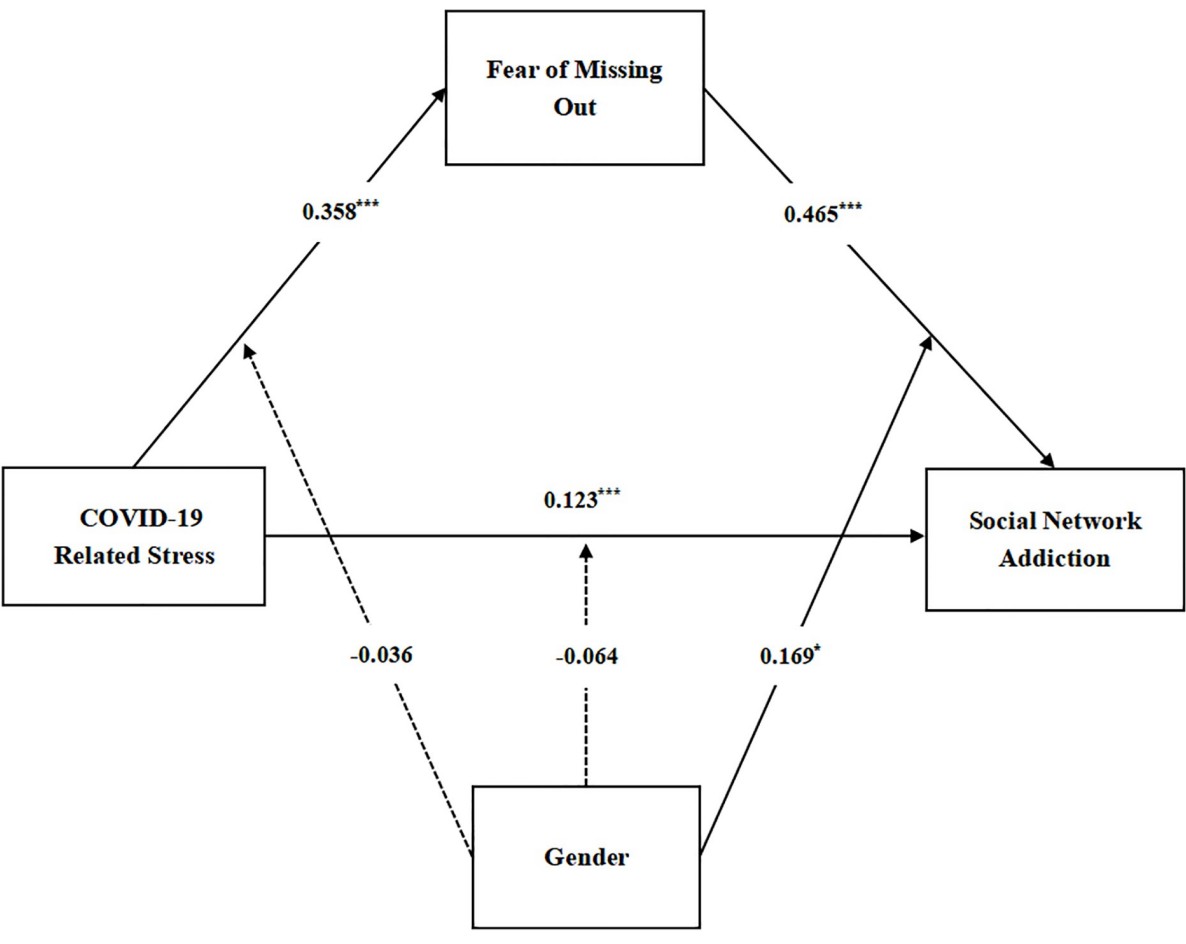

**Fig 2. Moderated mediation role of gender.**

in male college students (simple slope = 0.680, t = 10.323, p < 0.001) than females (simple slope = 0.490, t = 11.768, p < 0.001).

## Discussion

### Theoretical contributions

First, a significant positive effect of COVIID-19 related stress on SNA was observed in Chinese college students and H1 was supported, which is similar to previous studies concluding a significant predictive effect of stress on SNA [24, 26, 36, 69]. During the COVID-19 pandemic, a great deal of offline teaching and learning moved from offline to online [39], their lifestyle changed dramatically and socialization was restricted and relied on SNSs [105]. Excessive social network use may trigger SNA [14, 19]. In addition, previous studies have indicated that stress under specific conditions is one of the drivers of SNA [4, 36]. Whereas COVID-19 pandemic is a particular social environment, different aspects of stress associated may be faced by college students [39, 105, 106]. Social networks are often used as a channel to relieve the reality of stress [105, 107], particularly under COVID-19 pandemic. These reasons may contribute to the excessive use of social networks among college students during Covid-19 pandemic [64] and may result in SNA [1, 19, 36].

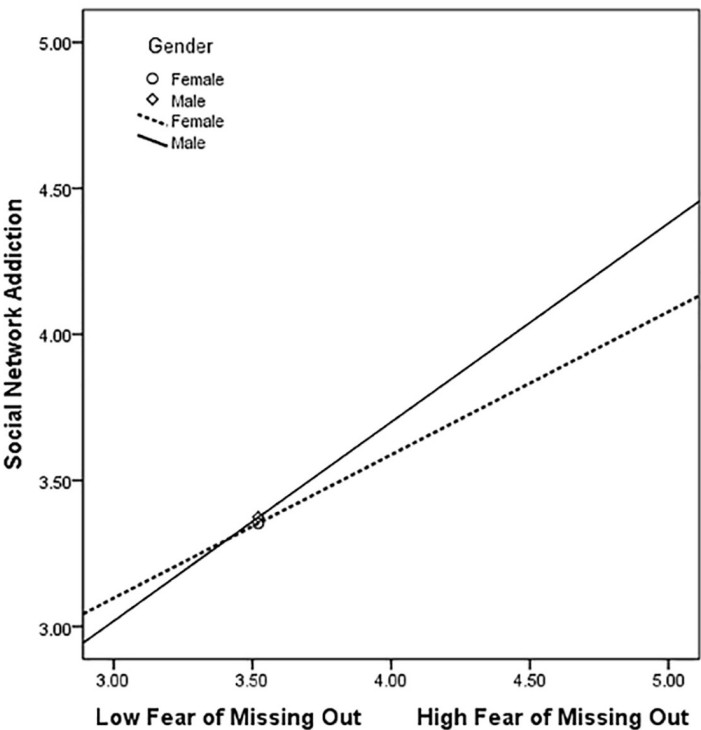

**Fig 3. Moderation effect of gender between FoMO and SNA.**

Second, the findings revealed that FoMO has a complementary partial mediation role between CRS and SNA. Then the H2 was supported. Particularly, CRS not only directly affects SNA but also indirectly influences SNA through FoMO, which is a state of anxiety with unmet social relationship needs [70] is a negative emotional state [44]. The COVID-19 pandemic has brought many restrictions to the life and study of college students, making them unable to get adequate social contact [94], as well as multi-faceted pressure [39], which will largely cause such negative emotions (FOMO) among college students. The FoMO, in turn, drove college students to use social networks frequently to satisfy social connections and engagement [74], increasing their psychological dependence on social networks [17] and leading to excessive use of social networks to the point of SNA [1, 36]. This study further extended the results of the above studies and supported the SCT. Particularly, the mediating role of FoMO was identified, and a complementary mediating model was constructed and verified by considering COVID-19 stress on college students as an environmental factor, FoMO as a personal factor, and SNA as a behavioral factor. It indicates that FoMO is a critical facilitator for SNA among college students and can complement COVID-19 related stress to increase their SNA.

Finally, notably, gender has a significant moderating effect only in the second half of the mediation model constructed in this study; the moderating effect in the first half of the path and the direct path did not reach statistical significance. Particularly, gender moderated the relationship between FoMO and SNA among college students, and the effect of FoMO on SNA was stronger among male college students than females. Thus H3b was supported. The findings are consistent with those of Koh & Kim's study [108]. In addition, in this study, gender had no significant moderating effect on the impact path of COVID-19 related stress on FoMO and SNA, thus H3a and H3c were not supported. The reason may be concluding no significant difference in the perception of stress by gender in a specific context of major public

health disasters (e.g., severe acute respiratory syndrome [SARS]) [109, 110]. Similar to SARS, COVID-19 is also an emergency and major public health disaster [38, 109]. Therefore, under the impact of such emergency and significant public health events, gender perception of pressure is not significantly different.

The significant moderating role of gender played in the effect of FoMO on SNA may be because of the following: First, according to gender social role differences, traditional male social roles require them to shoulder more responsibilities for obtaining survival resources. Although the current social situation has changed, such stereotyped thinking still exists in the current social culture [111], and this traditional concept of social role makes men's sense of crisis for social competition more intense [112]. This may lead men to worry more than women that others are having rewarding experiences that they themselves are not, and that they therefore have a greater need for constant attention and connection with others. Second, from the perspective of the status characteristics theory, men typically occupy a stronger status position than women; therefore, they seem to socially embody more purpose-oriented characteristics and make more decisions than women [113]. Therefore, men may need more information than women to help them make the right decisions to maintain their dominance [111]. Hence, men may be more afraid than women about missing out on helpful information or experiences, which may make men more susceptible to FoMO. Third, the number of deaths during COVID-19 was higher in men than in women [114], which may also make men more afraid of missing out on beneficial health-related messages. Fourth, males are more inclined to maintain their individuality socially [115]. SNSs, a private environment free from parental supervision, allow college students to make their identities in an unfettered environment [10], which may be more attractive to male students. Fifth, males appear to have a more apparent risk-taking attitude than females [116], wherein males are more likely to post personal information about themselves, such as phone numbers and addresses, on social network profiles, whereas females have greater privacy concerns and avoid disclosing identity information [116], this may also make men feel more comfortable indulging in social networking in a state of FOMO. In summary, all of these reasons are likely to make the effect of FoMO on social network addiction stronger among male college students than females. This result is an essential contribution of this study, and further enhances the understanding of gender differences among college students between FoMO and SNA.

## Practical contributions

This study provides some practical contributions:

First, in the future, when facing public health emergencies such as COVID-19, college students should actively make corresponding self-adjustment and actively participate in stress training courses to relieve the pressure brought by major public health disasters. Second, in the case of disasters such as COVID-19 that threaten public health, college students, especially male college students, should actively learn about FoMO, so as to correctly understand the concept and function of FoMO and make appropriate self-adjustment. Third, with the development of The Times, the difference between gender roles in current society has changed dramatically. Women may gradually have the same social power and shoulder the same social responsibilities as men [117] Male college students should actively change their thinking, abandon the traditional thinking of social roles, and adjust themselves to reduce their excessive sense of crisis.

## Conclusions

Overall, this study focused on some problems due to the COVID-19. We explored the effects of CRS on SNA, the mediating role of the FoMO, and the moderating role of gender. We

found that CRS not only significantly and positively predicted SNA of college students in the direct path but also significantly and positively influenced it through the mediator of FoMO in the indirect path. At the same time, this indirect effect was moderated by gender. Particularly, gender moderated the relationship between FoMO and SNA, and FoMO was a stronger predictor of SNA in male college students than in females. These findings support SCT and gender differences and broaden our understanding of the combined effects of COVID-19 pandemic and other major public health disasters on college students.

## Limitations and future research directions

There are still some limitations in this study. This study only conducted a questionnaire survey among college students in Yunnan Province, China, who were severely affected by the epidemic. Further research should consider sampling additional Chinese provinces. Second, this study was conducted on college students from China. Future studies could be conducted cross-culturally to compare the differences between Chinese and Western college students in the relationship between CRS and SNA or between different groups to expand the general applicability of the findings. Third, this study used a cross-sectional design and causal relationships between variables could not be inferred from the findings. Thus, longitudinal or experimental studies should be considered in future research.

## Supporting information

**S1 Checklist. STROBE statement—Checklist of items that should be included in reports of observational studies.**
(DOCX)

## Acknowledgments

Thanks to all the participants in this study.

## Author Contributions

**Conceptualization:** Ziao Hu, Jun Li, Maozheng Fu.

**Data curation:** Yangli Zhu, Jiafu Liu, Maozheng Fu.

**Formal analysis:** Ziao Hu.

**Investigation:** Yangli Zhu, Jiafu Liu, Maozheng Fu.

**Methodology:** Yangli Zhu, Jun Li.

**Supervision:** Jun Li.

**Writing – original draft:** Ziao Hu, Jun Li, Jiafu Liu.

**Writing – review & editing:** Ziao Hu, Jun Li.

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
