## [Decision Letter · Decision Letter 0]

12 May 2023

PONE-D-23-09029The COVID-19 Related Stress and Social Network Addiction among Chinese College Students: A Moderated Mediation ModelPLOS ONE

Dear Dr. Li,

Thank you for submitting your manuscript to PLOS ONE. After careful consideration, we feel that it has merit but does not fully meet PLOS ONE’s publication criteria as it currently stands. Therefore, we invite you to submit a revised version of the manuscript that addresses the points raised during the review process.

We look forward to receiving your revised manuscript.

Kind regards,

Alejandro Vega-Muñoz, Ph.D.

Academic Editor

PLOS ONE

Journal Requirements:

Reviewers' comments:

Reviewer's Responses to Questions

**Comments to the Author**

1. Is the manuscript technically sound, and do the data support the conclusions?

Reviewer #1: Yes

Reviewer #2: Partly

2. Has the statistical analysis been performed appropriately and rigorously? 

Reviewer #1: Yes

Reviewer #2: N/A

3. Have the authors made all data underlying the findings in their manuscript fully available?

Reviewer #1: Yes

Reviewer #2: Yes

4. Is the manuscript presented in an intelligible fashion and written in standard English?

Reviewer #1: Yes

Reviewer #2: Yes

5. Review Comments to the Author

Reviewer #1: The authors presented a study on the topic: “The COVID-19 Related Stress and Social Network Addiction among Chinese College Students: A Moderated Mediation Model” which examines the relationship between COVID-19 related stress (CRS) and social network addiction (SNA) among Chinese university students. In this study, the authors adopted a social cognitive theory and gender differences as determining factors in their theoretical framework. Both factors investigate the mediating role of fear of missing out (FoMO) as well as the moderating role of gender. There are few comments I wish to address as follows:

1. The hypothetical problem is not very clear; I would suggest that the authors outline their hypothesis by identifying the hypothesis and stating them in order. Authors can use bulletins to do that.

2. Authors should relate their findings according to the stated hypothesis.

3. The tables in the article should be properly drawn. Some lines are hidden with text hovering over the lines.

4. According to the gender influence in the study, can the authors provide an explanation or theoretical justification for why males might be more affected by FoMO compared to females.

Reviewer #2: Hypotheses 2 and 3, when speaking of a "mediating or moderating role", it is not clear in what sense. These hypotheses should be stated more concretely.

In the method, the steps to be followed for the validation of the instruments and the rest of the analysis should be indicated more clearly and in a logical order. The presentation of results should also follow this order, keeping the concordance between each item of the research.

In the practical contributions, the first part emphasizes Covid. A broader context should be given based on the current situation, which is no longer a health emergency. On the other hand, emphasis is placed on the actions to be taken by university teachers, however, the study does not focus on them; in this regard, the practical contributions should be related to the main line of research of this study.

6. PLOS authors have the option to publish the peer review history of their article (what does this mean?). If published, this will include your full peer review and any attached files.

Reviewer #1: No

Reviewer #2: No

---

## [Author Response · Author response to Decision Letter 0]

23 Jun 2023

Response to Reviewer 1 Comments

Dear reviewer 1:

Thank you very much for your time involved in reviewing the manuscript and your comments have further improved the quality of the manuscript.

We have carefully reviewed the comments and revised the manuscript accordingly. The modified section was already highlighted in yellow. Hope the explanation has fully addressed all of your concerns. Point-by-point response to reviewer are attached below this letter.

Please see the attachment.

Point 1: The hypothetical problem is not very clear; I would suggest that the authors outline their hypothesis by identifying the hypothesis and stating them in order. Authors can use bulletins to do that. 

Response 1: Thank you very much for your comments. We have presented the hypotheses in order and use bulletins to present the research hypotheses. Please see the lines 230-244 of our manuscript for detailed revisions.

Point 2: Authors should relate their findings according to the stated hypothesis.

Response 2: Thank you very much for your advice. We have linked research hypotheses with findings. Please see the lines 394, 399, 420, 423, 427, 450, 463, 483 and 485 of our manuscript for detailed revisions.

Point 3: The tables in the article should be properly drawn. Some lines are hidden with text hovering over the lines.

Response 3: Thanks so much for your reminding, we have adapted the tables. Please see the lines 369, 388, 409 and 428 of our manuscript for detailed revisions.

Point 4: According to the gender influence in the study, can the authors provide an explanation or theoretical justification for why males might be more affected by FoMO compared to females.

Response 4: We have added gender social role differences in the discussion section to explain why males were more affected by FoMO than women. Please see the lines 493-504 of the manuscript.

Response to Reviewer 2 Comments

Dear reviewer 2:

Thank you very much for your time involved in reviewing the manuscript and your comments have further improved the quality of the manuscript.

We have carefully reviewed the comments and revised the manuscript accordingly. The modified section was already highlighted in yellow. Hope the explanation has fully addressed all of your concerns. Point-by-point response to reviewer are attached below this letter.

Please see the attachment.

Point 1: Hypotheses 2 and 3, when speaking of a "mediating or moderating role", it is not clear in what sense. These hypotheses should be stated more concretely.

Response 1: Thanks for your advice, we have revised the hypothesis to make the hypothesis reflect more concretely. Please see the lines 166-168, 201-203 and 227-244 of our manuscript for specific modifications.

Point 2: In the method, the steps to be followed for the validation of the instruments and the rest of the analysis should be indicated more clearly and in a logical order. The presentation of results should also follow this order, keeping the concordance between each item of the research.

Response 2: According to your comments, we have modified the data analysis strategy in the "Statistical analysis" section to correspond to the sequence of results presentation. Please see the lines 295-334 of our manuscript for specific revisions.

Point 3: In the practical contributions, the first part emphasizes Covid. A broader context should be given based on the current situation, which is no longer a health emergency. On the other hand, emphasis is placed on the actions to be taken by university teachers, however, the study does not focus on them; in this regard, the practical contributions should be related to the main line of research of this study.

Response 3: Thank you for your comments. A broader context in the practice contributions has been given based on your recommendations and we shifted our focus from university teachers to the main line of research of this study (college students ). Please see the lines 523-533 of the manuscript.

---

## [Decision Letter · Decision Letter 1]

11 Aug 2023

The COVID-19 Related Stress and Social Network Addiction among Chinese College Students: A Moderated Mediation Model

PONE-D-23-09029R1

Dear Dr. Li,

We’re pleased to inform you that your manuscript has been judged scientifically suitable for publication and will be formally accepted for publication once it meets all outstanding technical requirements.

Kind regards,

Alejandro Vega-Muñoz, Ph.D.

Academic Editor

PLOS ONE

Additional Editor Comments (optional):

Both reviewers have accepted the manuscript.

Reviewers' comments:

Reviewer's Responses to Questions

**Comments to the Author**

1. If the authors have adequately addressed your comments raised in a previous round of review and you feel that this manuscript is now acceptable for publication, you may indicate that here to bypass the “Comments to the Author” section, enter your conflict of interest statement in the “Confidential to Editor” section, and submit your "Accept" recommendation.

Reviewer #1: All comments have been addressed

Reviewer #2: All comments have been addressed

2. Is the manuscript technically sound, and do the data support the conclusions?

Reviewer #1: Yes

Reviewer #2: Yes

3. Has the statistical analysis been performed appropriately and rigorously? 

Reviewer #1: Yes

Reviewer #2: Yes

4. Have the authors made all data underlying the findings in their manuscript fully available?

Reviewer #1: Yes

Reviewer #2: Yes

5. Is the manuscript presented in an intelligible fashion and written in standard English?

Reviewer #1: Yes

Reviewer #2: Yes

6. Review Comments to the Author

Reviewer #1: The publication exhibits a remarkable and sophisticated writing style, showcasing the mastery of English language construction. The study was meticulously conducted, reflecting a commendable commitment to adhering to research ethics and best practices. The authors demonstrated a clear understanding of ethical considerations, ensuring the well-being and rights of participants throughout the research process.

The comprehensive nature of this study is evident, with the authors leaving no stone unturned in their pursuit of knowledge and understanding. The methodology employed was sound, fostering confidence in the validity and reliability of the results obtained. The data analysis was both robust and thorough, leading to meaningful insights and valuable contributions to the field of study.

It is evident that the authors have taken great care in crafting their publications and presenting their findings in a logical and coherent manner. The clarity of their arguments and the organization of the content make it an engaging read for both experts and non-experts in the subject matter.

Given the rigour and high quality of this research, the publication is undoubtedly suitable for inclusion in this esteemed journal. Its significance and relevance to the broader academic community make it a valuable addition to the body of literature in the field.

While the publication meets the standards for this journal, it is always beneficial to invite external peer reviews to further validate the research and provide constructive feedback. This process can strengthen the credibility of the study and ensure that potential blind spots or biases are addressed.

Additionally, I would like to commend the authors for their adherence to research ethics and responsible conduct. Ethical considerations are of utmost importance in research, and their conscientious approach provides a positive example for others in the academic community.

Finally, I suggest that the authors should consider sharing their research data openly, if possible, to promote transparency and facilitate reproducibility. This practice not only benefits the research community but also upholds the principles of scientific integrity.

Overall, the publication is a testament to the author's dedication, expertise, and adherence to research ethics, making it a worthy addition to the journal's prestigious collection of scholarly works.

Reviewer #2: The authors have answered the questions of the first round of review, making the required adjustments in the article.

7. PLOS authors have the option to publish the peer review history of their article (what does this mean?). If published, this will include your full peer review and any attached files.

Reviewer #1: No

Reviewer #2: No

---

## [Editor Report · Acceptance letter]

16 Aug 2023

PONE-D-23-09029R1 

The COVID-19 Related Stress and Social Network Addiction among Chinese College Students: A Moderated Mediation Model 

Dear Dr. Li:

I'm pleased to inform you that your manuscript has been deemed suitable for publication in PLOS ONE. Congratulations! Your manuscript is now with our production department. 

Kind regards, 

on behalf of

Dr. Alejandro Vega-Muñoz 

Academic Editor

PLOS ONE